# Computational Comparison of Differential Splicing Tools for Targeted RNA Long-Amplicon Sequencing (rLAS)

**DOI:** 10.3390/ijms26073220

**Published:** 2025-03-30

**Authors:** Hiroki Ura, Hisayo Hatanaka, Sumihito Togi, Yo Niida

**Affiliations:** 1Center for Clinical Genomics, Kanazawa Medical University Hospital, 1-1 Daigaku, Uchinada, Kahoku 920-0923, Ishikawa, Japantogi@kanazawa-med.ac.jp (S.T.); niida@kanazawa-med.ac.jp (Y.N.); 2Division of Genomic Medicine, Department of Advanced Medicine, Medical Research Institute, Kanazawa Medical University, 1-1 Daigaku, Uchinada, Kahoku 920-0923, Ishikawa, Japan

**Keywords:** splicing analysis, splicing tools, RNA-Seq, next-generation sequencing (NGS), genetic diagnosis

## Abstract

RNA sequencing (RNA-Seq) is a powerful technique for the quantification of transcripts and the analysis of alternative splicing. Previously, our laboratory developed the targeted RNA long-amplicon sequencing (rLAS) method, which has the advantage of allowing deep analysis of targeted specific transcripts. The computational tools for analyzing RNA-Seq data have boosted alternative splicing research by detecting and quantifying splicing events. However, the performance of these splicing tools has not yet been investigated for rLAS. Here, we evaluated the performance of four splicing tools (MAJIQ, rMATS, MISO, and SplAdder) using samples with different types of known splicing events (exon-skipping, multiple-exon-skipping, alternative 5′ splicing, and alternative 3′ splicing). MAJIQ was able to detect all of the types of events, but it was unable to detect one of the exon-skipping events. On the other hand, rMATS was able to detect all of the exon-skipping events. However, rMATS failed to detect other types of events besides exon-skipping events. Both MISO and SplAdder were unable to detect any of the events. These results indicate that MAJIQ presents better performance for the different types of splicing events in rLAS and that rMATS shows better performance for exon-skipping splicing events.

## 1. Introduction

Transcriptome analysis through RNA sequencing (RNA-Seq) is a powerful tool for genome-wide quantification of RNA expression and for the detection of novel and known splicing events in the expressing transcripts [1,2,3,4]. Alternative splicing is recognized as an important intracellular event in the post-transcriptional regulation of gene splicing [5,6]. Genome-wide studies estimate that about 90% of human genes undergo some level of alternative splicing [6,7]. The types of alternative splicing include exon skipping, multiple exon skipping, alternative 5′ splicing, and alternative 3′ splicing [8,9] (Figure 1A). Exon skipping occurs when an exon is spliced out together with its flanking introns, and similarly, multiple exon skipping occurs when more than two exons are spliced out. Alternative 5′ splicing or alternative 3′ splicing is caused by alternative splice sites that may result in the inclusion (gain) and/or exclusion (loss) of alternatively spliced regions. Alternative splicing occurs simultaneously in multiple genes during development and cell differentiation [10,11,12,13,14]. Since abnormal alternative splicing has been reported in congenital and acquired human diseases, including cancers, abnormal alternative splicing events may be new tools for disease diagnosis and classification [15,16].

Over the last few decades, the widespread use of next-generation sequencing (NGS) technologies has enabled the detection of genetic variants within the genome that represent germline mutations or somatic mutations in terms of genetic diagnosis. Although it is possible to predict the variants within the exon, it is difficult to predict the pathogenicity of the variants within the intron. To assess the pathogenicity of the intron variants, it is necessary to investigate whether the intron variants cause abnormal alternative splicing. Recently, several RNA-Seq methods have been developed [17], and most RNA-Seq methods focus on genome-wide quantification of RNA expression. Therefore, these RNA-Seq methods are not suitable for fine-tuned alternative splicing analysis of a specific gene, as it is possible that the lower depth is in lower-expressing genes. To resolve this issue, our laboratory developed the targeted RNA long-amplicon sequencing (rLAS) method [18] (Figure 1B). The rLAS method has enabled the detection of alternative splicing events in low-expressing genes using targeted specific transcript amplification. In addition, it makes it possible to analyze long transcripts because the rLAS method is able to stably amplify the long-amplicon from full-length double-stranded cDNA synthesized by the SMARTer method. In genetic diagnosis, it is necessary to detect abnormal splicing events in specific disease-causing genes. The rLAS method is a more cost-effective and accurate method compared to RNA-Seq because it focuses on the specific disease-causing genes rather than other comprehensive genes. However, in rLAS, abnormal splicing is manually extracted from the data visualized by Integrative Genomic Viewer (IGV) and is not performed by software.

With the recent advances in RNA-Seq, whole-transcriptome analysis of not only quantification but also alternative splicing has become feasible, requiring the development of computational tools to accurately detect alternative splicing events. Over the past decade, many computational alternative splicing tools have been developed and evaluated, many of which have the ability to accurately detect and quantify alternative splicing events [19]. However, the performance of these splicing tools has not yet been investigated for targeted RNA-Seq methods such as rLAS. Here, we evaluate four commonly used alternative splicing tools (MAJIQ [20], rMATS [21], MISO [22], and SplAdder [23]) using samples with different types of known splicing events (exon-skipping, multiple-exon-skipping, alternative 5′ splicing, and alternative 3′ splicing) (Table 1, Figure 1C). These known splicing events were previously identified in our laboratory. Three known exon-skipping variants were identified in a patient with fumarate hydratase deficiency carrying an intron variant in the *FH* gene, a patient with anhidrotic ectodermal dysplasia carrying an intron variant in the *EDA* gene, and a patient with epidermodysplasia verruciformis carrying an intron variant in the *TMC8* gene [24]. The known multiple-exon-skipping variant was identified in a patient with tuberous sclerosis complex carrying a large deletion in the *TSC2* gene. The known alternative 5′ splicing variant was identified in a patient with hereditary hemorrhagic telangiectasia carrying an intron variant in the *ENG* gene. The known alternative 3′ splicing variant was identified in siblings, patients with X-linked Ohdo syndrome carrying an intron variant in the *MED12* gene [25,26]. The mapping data are required for splicing analysis using these computational splicing tools. Many mapping tools have been developed and evaluated to accurately quantify the transcriptome in RNA-Seq [27]. However, the performance of a combination of the splicing tools and mapping tools in rLAS has not yet been evaluated. In this study, we compare the performance of two well-known mapping tools (HISAT2 [28] and STAR [29]) combined with the splicing tools (Figure 1C).

## 2. Results

### 2.1. Comparison Between HISAT2 and STAR for Targeted RNA Long-Amplicon Sequencing (rLAS) Mapping

To evaluate the performance of the mapping tools HISAT2 and STAR for targeted RNA long-amplicon sequencing (rLAS) mapping, we compared the mapping rate (Figure 2A,B). The mapping rate of HISAT2 was slightly better than that of STAR, except for *ENG* (Figure 2A). Also, the on-target rate of HISAT2 was slightly better than that of STAR, except for *EDA*. However, the difference in the on-target rate was much smaller than the mapping rate, indicating that off-target mapping is slightly increased in HISAT2, but there is almost no difference between HISAT2 and STAR for rLAS mapping. The coverage of HISAT2 and STAR was uniform in the transcripts (Figure 2C). These results indicated that the performance of HISAT2 and STAR for the mapping of rLAS is almost the same, although there are slight differences.

### 2.2. Computational Comparison of Targeted RNA Long-Amplicon Sequencing (rLAS) for Exon-Skipping Variants

To evaluate the performance of the splicing tools MAJIQ, rMATS, MISO, and SplAdder for the exon-skipping variant, we performed the rLAS on three patient samples with known exon-skipping variants (Figure 3A–C): the patient with the known exon-skipping variant in the *FH* gene that causes exon 2 skipping, NC_000001.11(NM_000143.4):c.267+1G>A (Figure 3A); the patient with the exon 6-skipping variant in the *EDA* gene on the X chromosome, NC_000023.11(NM_001399.5):c. 793+3A>C (Figure 3B); and the patient with the exon 3-skipping variant in the *TMC8* gene, NC_000017.11(NM_152468.5):c.298+1G>A (Figure 3C). The average depth in the target region of the rLAS for HISAT2 and STAR was about 2300 in *FH*, about 1000 in *EDA*, and about 3000 in the *TMC8* gene (Figure 3D). In the *FH* gene, MAJIQ and rMATS detected an exon-skipping splicing event (Figure 3E). On the other hand, MISO and SplAdder did not detect any splicing events. In the *EDA* gene, all the splicing tools, MAJIQ, rMATS, MISO, and SplAdder, detected an alternative 5′ splicing event (Figure 3F). However, only MAJIQ and rMATS detected an exon-skipping splicing event. In the *TMC8* gene, only MAJIQ and rMATS detected the exon-skipping splicing events, although MISO detected an alternative 5′ splicing event and an alternative 3′ splicing event (Figure 3G). HISAT2 and STAR are almost the same as rLAS in terms of the detected splicing events, although there are slight differences. Both MAJIQ and rMATS were able to detect the known exon-skipping splicing events in the *FH* and *TMC8* genes (Figure 3H). However, only rMATS was able to detect the known exon-skipping event in the *EDA* gene. On the other hand, MISO and SplAdder were unable to detect the known exon-skipping splicing events in any of the genes. Although the percent spliced in (PSI) of the known exon-skipping splicing events in the *FH* gene is lower than others in STAR for rMATS, the PSI is almost the same for the other known exon-skipping splicing events (Figure 3I). These results indicated that MAJIQ and rMATS are suitable for the analysis of exon-skipping splicing events in rLAS. rMATS is better than MAJIQ at detecting exon-skipping splicing events, but there is a difference in PSI calculation in rMATS.

### 2.3. Computational Comparison of Targeted RNA Long-Amplicon Sequencing (rLAS) for Multiple-Exon-Skipping Variant

To evaluate the performance of the splicing tools MAJIQ, rMATS, MISO, and SplAdder for the multiple-exon-skipping variant, we performed the rLAS on the tuberous sclerosis complex (TSC) patient sample with the known multiple-exon-skipping variant in the *TSC2* gene. The TSC patient with the known multiple-exon-skipping variant has the mutation (NC_000016.:g.2056989_2074645del) in the *TSC2* gene that causes the multiple-exon skipping (Figure 4A). The average depth on the target region of the rLAS for HISAT2 and STAR was about 7000 in the *TSC2* gene (Figure 4B). Not only MAJIQ but also rMATS, MISO, and SplAdder detected the splicing events except the multiple-exon-skipping event (Figure 4C). However, only MAJIQ was able to detect the multiple-exon-skipping events in rLAS. Although HISAT2 and STAR in MAJIQ detected the multiple-exon-skipping events, only HISAT2 in MAJIQ was able to detect the known multiple-exon-skipping event (Figure 4D). These results indicated that only the combination of HISAT2 and MAJIQ is suitable for the analysis of multiple-exon-skipping splicing events in rLAS.

### 2.4. Computational Comparison of Targeted RNA Long-Amplicon Sequencing (rLAS) for the Alternative 5′ Splicing Variant and Alternative 3′ Splicing Variant

To evaluate the performance of the splicing tools MAJIQ, rMATS, MISO, and SplAdder for the exon-skipping variant, we performed the rLAS on three patient samples with known alternative 5′ and 3′ splicing variants (Figure 5A,B). The patient with the known alternative 5′ splicing variant in the *ENG* gene that causes alternative 5′ splicing, NC_000009.12(NM_000118.4): c.1311+172C>G (Figure 5A). The siblings with the known alternative 3′ splicing variant in the *MED12* gene that causes alternative 3′ splicing, NC_000023.11(NM_005120.3): c.887G>A (Figure 5B). The average depth in the target region of the rLAS for HISAT2 and STAR was about 5500 in ENG and about 3000 in the *MED12* gene (Figure 5C). In the *ENG* gene, MAJIQ was able to detect the alternative 5′ splicing event (Figure 5D). Although rMATS detected the exon-skipping events, the alternative 5′ splicing event was not detected. On the other hand, no splicing events were detected by MISO and SplAdder in the *ENG* gene. In the *MED12* gene, MAJIQ and MISO were able to detect not only the alternative 3′ splicing event but also other splicing events (Figure 5E). rMATS detected the exon-skipping events but not the alternative 3′ splicing event in the *MED12* gene as well as in the ENG gene. No splicing events were detected by SplAdder in the *MED12* gene or in the *ENG* gene. Only MAJIQ was able to detect the known alternative 5′ splicing event in the *ENG* gene and the known alternative 3′ splicing event in the *MED12* gene (Figure 5F). Although MISO also detected the alternative 3′ splicing events, the known alternative 3′ splicing event was not included in the detected alternative 3′ splicing events. There were no differences between HISAT2 and STAR in terms of the detection of splicing events (Figure 5D,E). Also, the PSI of the known alternative 5′ splicing event and alternative 3′ splicing events were the same (Figure 5G). These results indicate that only MAJIQ is suitable for the analysis of alternative 5′ splicing events and alternative 3′ splicing events in rLAS.

### 2.5. Computational Comparison of Targeted RNA Long-Amplicon Sequencing (rLAS) for Read Count Limitation

To evaluate the performance of the splicing tools, MAJIQ, and rMATS in read count limitation, we extracted any number of reads using the SeqKit software (version 0.13.2). As to the known exon-skipping event in the *FH* gene, both HISAT2 and STAR were able to detect it in MAJIQ at 1/10 reads but not at 1/100 reads (Figure 6A). On the other hand, STAR, but not HISAT2, was able to detect it in rMATS at 1/100 reads as well as at 1/10 reads. Although there is almost the same value of PSI at 1/10 reads for both MAJIQ and rMATS, the value of PSI was decreased at 1/100 reads in rMATS. As to the known exon-skipping event in the *EDA* gene, both HISAT2 and STAR were able to detect it in rMATS at 1/10 reads but not at 1/100 reads, and the value of PSI was completely the same at 1/10 reads. (Figure 6B). As to the known exon-skipping event in the *TMC8* gene, both HISAT2 and STAR were able to detect it in both MAJIQ and rMATS at 1/10 reads but not at 1/100 reads (Figure 6C). There is almost the same value of PSI at 1/10 reads in both MAJIQ and rMATS. In the known multiple-exon-skipping event of TSC2, MAJIQ was not able to detect it even at 1/10 reads (Figure 6D). As to the known 5′ alternative splicing event in ENG, HISAT2 and STAR were able to detect it in MAJIQ at 1/10 reads and even at 1/100 reads (Figure 6E). Moreover, there is almost the same value of PSI at 1/10 reads and even at 1/100 reads in both HISAT2 and STAR. As to the known 3′ alternative splicing event in MED12 in patient 1, only HISAT2 was able to detect it in MAJIQ at 1/10 reads (Figure 6F). There is almost same value of PSI at 1/10 reads in patient 1. These results indicate that the number of reads required to detect the splicing event is different for each splicing event rather than each type of splicing event. It is possible to detect any splicing event with approximately 10,000 reads in rLAS.

## 3. Discussion

Next-generation sequencing (NGS)-based alternative splicing analysis can serve as a powerful tool for the detection and quantification of alternative splicing events, not only in developmental research but also in the genetic diagnostics of various human diseases. Recently, several RNA-Seq methods have been developed, and most RNA-Seq methods focus on genome-wide quantification of RNA expression. Therefore, these RNA-Seq methods are not suitable for alternative splicing analysis of a specific gene, as it is possible that the lowest depth is in lower-expressing genes. To address this issue, our laboratory has developed the targeted RNA long-amplicon sequencing (rLAS) method. rLAS has advantages for the detection of alternative splicing events in targeted genes due to its targeted specific transcript amplification. The rLAS method is the more cost-effective and accurate method compared to RNA-Seq because it focuses on the specific disease-causing genes rather than other, comprehensive genes. In genetic diagnosis, the rLAS method also provides advantages. Over the past decade, many computational alternative splicing tools have been developed and evaluated. However, it is not yet known which splicing tools are suitable for the rLAS method.

First, we evaluated the performance of the mapping tools (HISAT2 and STAR) because the mapping data are required for splicing analysis using these computational splicing tools. Although the mapping rate of HISAT2 was better than STAR, the on-target rate was almost the same between HISAT2 and STAR. Next, we evaluated the performance of the alternative splicing tools (MAJIQ, rMATS, MISO, and SplAdder) using samples with different types of the known splicing events (exon-skipping, multiple-exon-skipping, alternative 5′ splicing, and alternative 3′ splicing events). MAJIQ was able to detect all types of the known splicing events, with only one known exon-skipping event not detected. rMATS was able to detect all known exon-skipping splicing events but not known multiple-exon-skipping, alternative 5′ splicing, or alternative 3′ splicing events. On the other hand, MISO and SplAdder were unable to detect any of the known splicing events in rLAS. These results indicate that MAJIQ and rMATS are suitable for the rLAS method and that it is possible to detect all splicing events using both MAJIQ and rMATS in rLAS. rMATS with STAR was able to detect the known skipping splicing event in the FH gene at a depth of about 20, and MAJIQ was also able to detect the known alternative 5′ splicing event in the ENG gene at a depth of about 50. However, the detection sensitivity varies from splicing event to event, so a depth of around 10,000 appears to be required to reliably detect all types of splicing events. Although the performance of HISAT2 was almost better than STAR, the performance of the mapping tools also varied for different splicing events, indicating that one can detect all splicing events using both HISAT2 and STAR in rLAS.

In combination with the mapping tools (HISAT2 and STAR) and the alternative splicing tools (MAJIQ and rMATS), all known splicing events can be detected in rLAS. However, it is possible that other mapping tools (GSNAP [30], SpliceMap [31], Segemehl [32], and GEM [33]) and alternative splicing tools (SUPPA2 [34], Whippet [35]) exist that may be more suitable for rLAS. These mapping tools (HISAT2 and STAR) have different biases for different genes. Similarly, the alternative splicing tools (MAJIQ and rMATS) have different performance across different splicing events. Therefore, it is better to evaluate the performance of other mapping tools and other alternative splicing tools in the future. In this study, the samples from the seven patients with different known splicing events were used for rLAS. For a more comprehensive evaluation of rLAS, it is necessary to increase the number of samples with splicing events.

In this study, we used the Illumina short-read sequencer for rLAS to detect the splicing events, and in the future, rLAS may be applied to long-read sequencers such as Nanopore [36] and PacBio [37]. Using the rLAS method with a long-read sequencer will provide not only the individual splice event information but also all splice event information in a full-length transcript. Although the long-read sequencer can provide the information expressed in full-length transcripts, there is a cost required to analyze the low-expressing genes. On the other hand, the rLAS method provides low-cost genetic diagnosis by focusing on only the targeted disease-associated genes. However, the rLAS method may not identify the splicing events of other non-targeted transcripts of the disease-associated gene because the rLAS method focuses only on a targeted transcript that has the functional domain. In the future, the method of capturing all the transcripts of a target disease-associated gene using biotin-labelled specific exon probes and sequencing them using a long-read sequencer will provide all the splicing events in the target disease-associated gene.

## 4. Materials and Methods

### 4.1. Total RNA Extraction

The details of the procedure have been reported previously [38,39]. Briefly, total RNA from human induced pluripotent stem cells (iPSCs) was extracted with TRIzol reagent (Thermo Fisher Scientific, Waltham, MA, USA) according to the manufacturer’s instructions. The RNA concentration and purity were measured spectrophotometrically (Nanodrop). The RNA integrity number (RIN) indicates the degree of RNA degradation. RIN was measured by TapeStaion 4200 with RNA Screen Tape (Agilent Technologies, Santa Clara, CA, USA).

### 4.2. Targeted RNA Long-Amplicon Sequecning (rLAS)

The full-length double-stranded cDNA was synthesized from 50 ng of total RNA using SMART-Seq^®^ HT Kit (Takara Bio USA, Mountain View, CA, USA) according to the manufacturer’s standard protocols (Figure 1B). To amplify the double-stranded full-length cDNA of the target genes by long-range PCR, we designed the long-range PCR primers with Primer3 v.0.4.0 accessed on 4 July 2020 (http://bioinfo.ut.ee/primer3-0.4.0/, accessed on 26 March 2025) using the following parameters: primer length, 20–22–27 mer; Tm, 61.5–62.0–62.5 °C; max Tm difference, 0.1 °C; GC%, 45–50–60; and GC Clump, 2, with the other parameters being used with default settings (Table 2). Touch-down PCR cycles were used for the long-range PCR amplification with the KOD One enzyme (TOYOBO): 5 cycles of 98 °C for 10 s and 74 °C for 5 min, 5 cycles of 98 °C for 10 s and 72 °C for 5 min, 5 cycles of 98 °C for 10 s and 70 °C for 5 min, and 25 cycles of 98 °C for 10 s and 68 °C for 5 min. Each PCR reaction contained 1 μL of 1 ng/μL of the full-length double-stranded cDNA in a 10 μL reaction volume, and the final primer concentration was 0.15 μM.

### 4.3. Library Preparation and Sequencing

Targeted long-amplicons were purified using AMPure XP beads (Beckman Coulter Life Sciences, Indianapolis, IN, USA), and the libraries were prepared using a Nextera Flex DNA kit (Illumina, San Diego, CA, USA) according to manufacturer’s protocol. Libraries were quantified using an HS Qubit dsDNA assay (Thermo Fisher Scientific) and TapeStation 4200. The size distribution of the libraries was qualified on TapeStation 4200 using High-Sensitivity D1000 ScreenTape. A 12.5 pmol/L library was sequenced on an Illumina MiSeq system (2 × 250 cycles) according to the standard Illumina protocol (Illumina). The FASTQ files were generated using MiSeq Local Run Manager v3 (Illumina).

### 4.4. Data Analysis

The FASTQ files were aligned to the reference human genome (hg38) using HISAT2 (version 2.1.0) [28] or STAR (version 2.7.10b) [29]. Subsequently, the splicing tools, MAJIQ (version 2.5.1) [20], rMATS (version 4.1.1) [21], MISO (version 0.5.4) [22], and SplAdder (version 2.4.2) [23], were used with default parameter settings to detect the splicing events and calculate the PSI of the detected splicing events in rLAS. For analysis and interpretation, we used SAMtools (version 1.9) [40], SeqKit (version 0.13.2) [41], and RSeQC (version 3.0.1) [42]. For visualization, the Integrative Genomic Viewer (IGV) (version 2.11.9) [43] was used.

## 5. Conclusions

We evaluated the performance of four splicing tools (MAJIQ, rMATS, MISO, and SplAdder) using samples with different types of known splicing events (exon-skipping, multiple-exon-skipping, alternative 5′ splicing, and alternative 3′ splicing). MAJIQ was able to detect all types of the events but was unable to detect one of the exon-skipping events. On the other hand, rMATS was able to detect all of the exon-skipping events. However, rMATS failed to detect other types of events besides exon-skipping events. The results indicate that MAJIQ presents better performance for the different types of splicing events in rLAS and that rMATS shows better performance for the exon-skipping splicing events than MAJIQ.

## Figures and Tables

**Figure 1 ijms-26-03220-f001:**
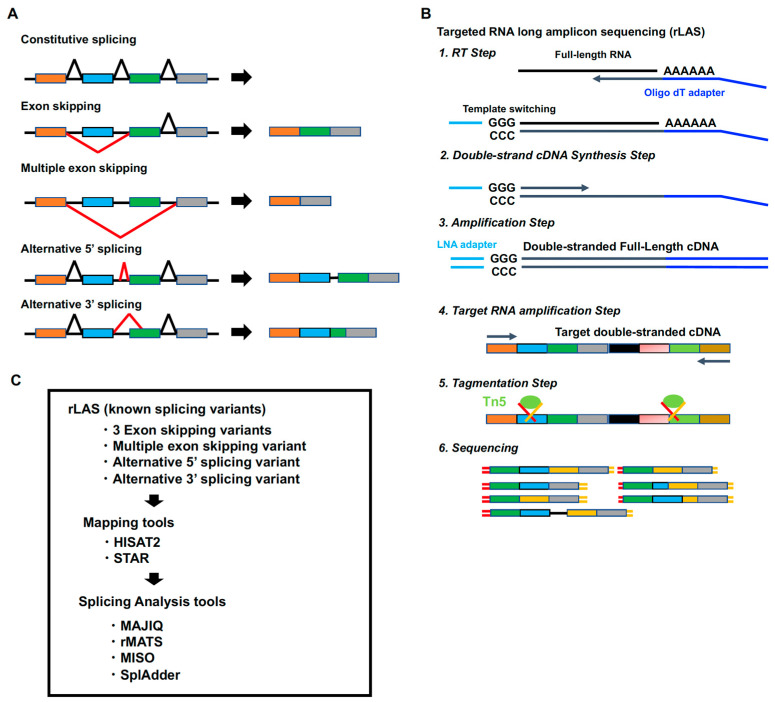
Workflow for targeted RNA long-amplicon sequencing (rLAS). (**A**) Alternative splicing types. (**B**) Workflow. (**C**) Evaluation scheme of rLAS.

**Figure 2 ijms-26-03220-f002:**
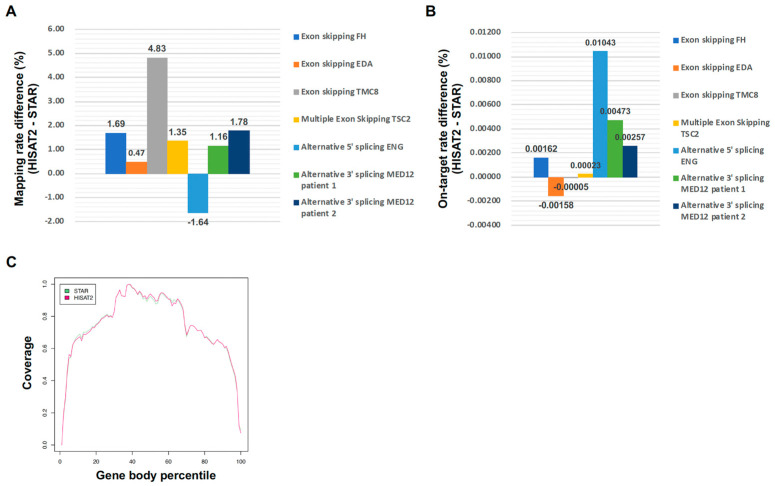
Comparison between HISAT2 and STAR. (**A**) Mapping rate difference between HISAT2 and STAR. (**B**) On-target rate difference between HISAT2 and STAR. (**C**) Gene body coverage.

**Figure 3 ijms-26-03220-f003:**
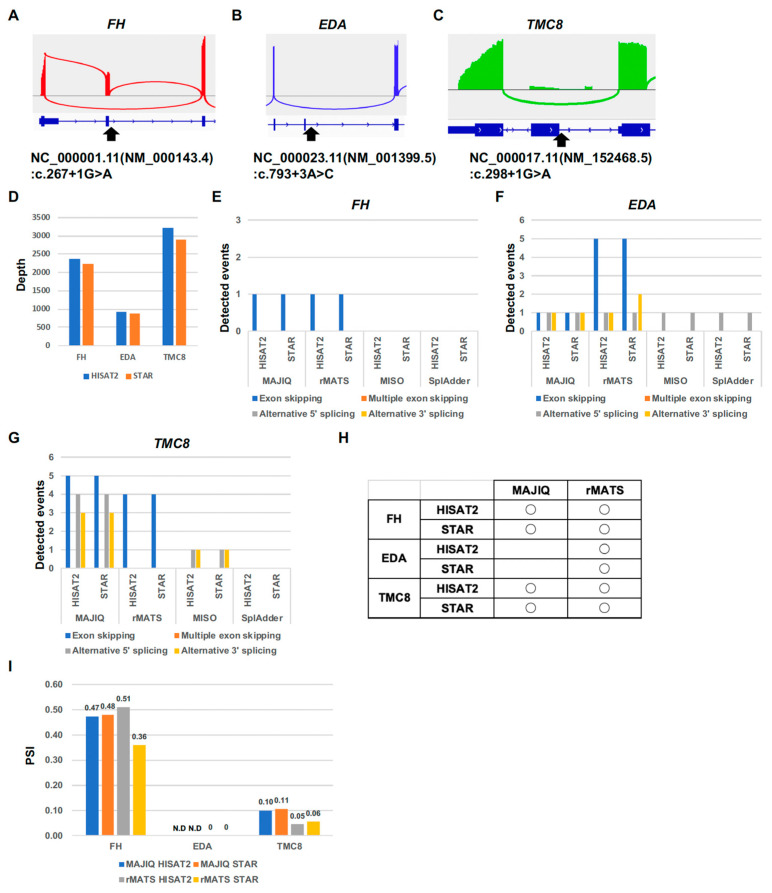
Comparison of splicing tools for exon-skipping events. (**A**) Sashimi plot of the known exon-skipping event in the FH gene. (**B**) Sashimi plot of the known exon-skipping event in the EDA gene. (**C**) Sashimi plot of the known exon-skipping event in the TMC8 gene. Arrowhead indicates the intron variant. The minimum junction coverage setting is 250. (**D**) Depth in the target gene. (**E**) Number of detected splicing events in the FH gene. (**F**) Number of detected splicing events in the EDA gene. (**G**) Number of detected splicing events in the TMC8 gene. (**H**) Table of the known exon-skipping events detected in MAJIQ and rMATS. (**I**) PSI of the known exon-skipping events in MAJIQ and rMATS.

**Figure 4 ijms-26-03220-f004:**
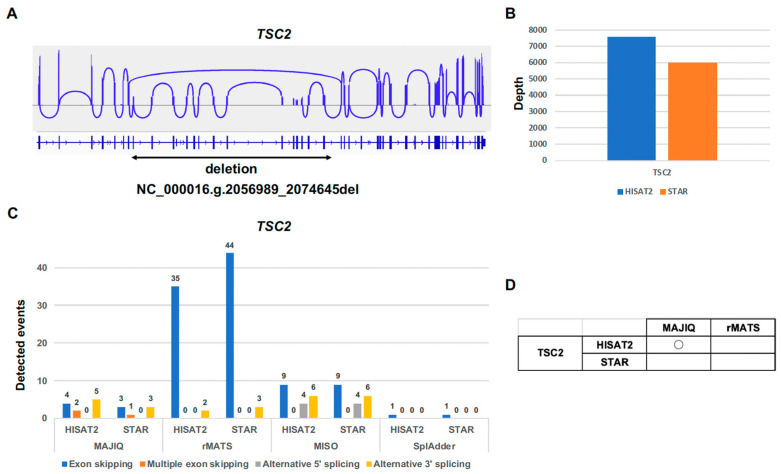
Comparison of splicing tools for multiple-exon-skipping event. (**A**) Sashimi plot of the known multiple-exon-skipping event in the TSC2 gene. The minimum junction coverage setting is 250. (**B**) Depth in the TSC2 gene. (**C**) Number of detected splicing events in the TSC2 gene. (**D**) Table of the known multiple-exon-skipping event detected in MAJIQ.

**Figure 5 ijms-26-03220-f005:**
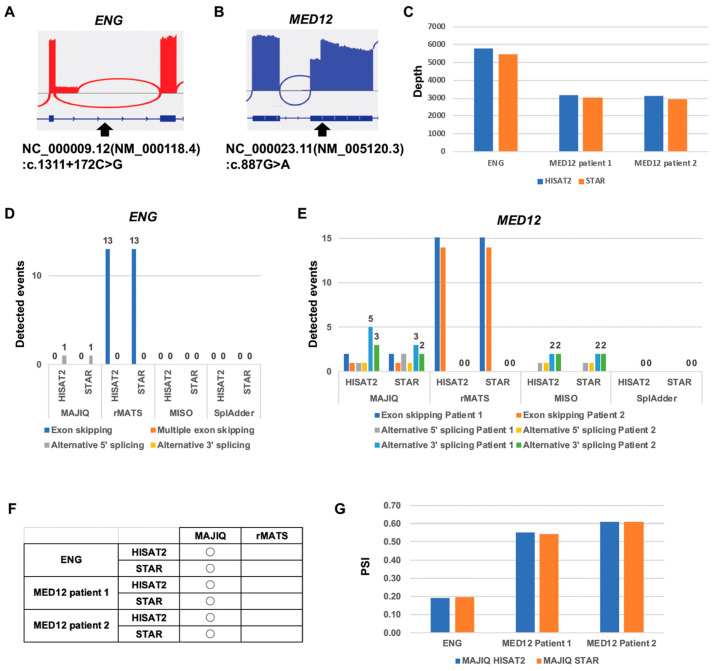
Comparison of splicing tools for alternative 5′ and 3′ splicing events. (**A**) Sashimi plot of the known alternative 5′ splicing event in the ENG gene. (**B**) Sashimi plot of the known alternative 3′ splicing event in the MED12 gene. The minimum junction coverage setting is 250. (**C**) Depth in the target gene. (**D**) Number of detected splicing events in the ENG gene. (**E**) Number of detected splicing events in the MED12 gene. (**F**) Table of the known alternative 5′ and alternative 3′ splicing events detected in MAJIQ and rMATS. (**G**) PSI of the known alternative 5′ and alternative 3′ splicing events in MAJIQ.

**Figure 6 ijms-26-03220-f006:**
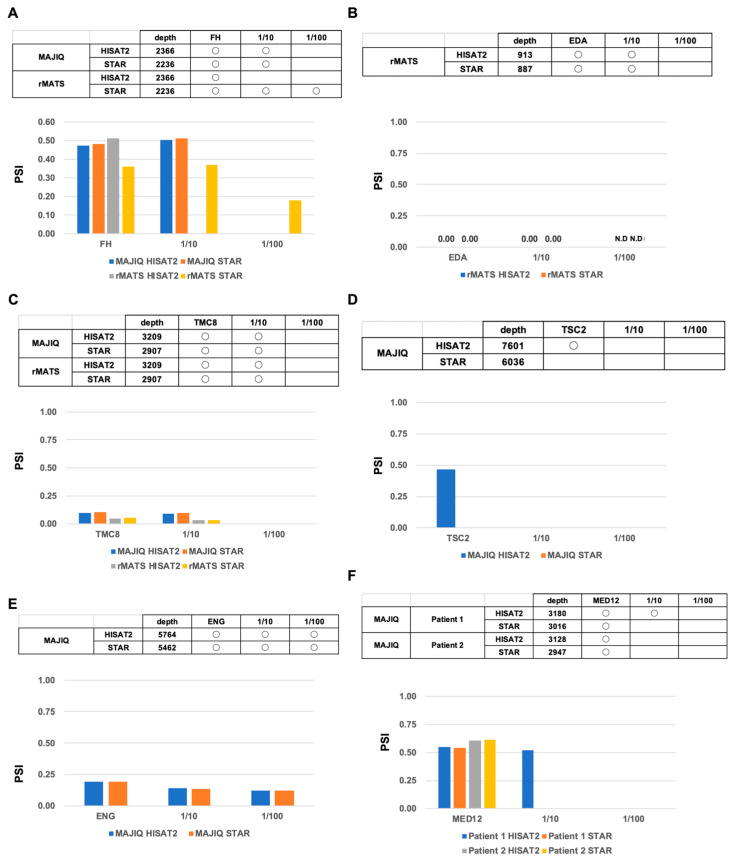
Comparison of splicing tools in a limited number of reads. (**A**) Table and PSI of the known exon-skipping splicing events in the FH gene detected by MAJIQ and rMATS. (**B**) Table and PSI of the known exon-skipping splicing events in the EDA gene detected by rMATS. (**C**) Table and PSI of the known exon-skipping splicing events in the TMC8 gene detected by MAJIQ and rMATS. (**D**) Table and PSI of the known multiple-exon-skipping splicing events in the TSC2 gene detected by MAJIQ. (**E**) Table and PSI of the known alternative 5′ splicing events in the ENG gene detected by MAJIQ. (**F**) Table and PSI of the known alternative 3′ splicing events in the MED12 gene detected by MAJIQ and rMATS.

**Table 1 ijms-26-03220-t001:** List of variants analyzed in this study.

Variant Type	Gene	dbSNP_ID	Reference Sequence	DNA Level (hg38)	cDNA Level	RNA Level	Protein Level
Exon skipping	*FH*	rs878853691	NC_000001.11 (NM_000143.4)	g.241517181 C>T	c.267+1G>A	r.133_267del (exon 2 skip)	p.(Ala45_Pro89del)
	*EDA*	NA	NC_000023.11 (NM_001399.5)	g.70030523 A>C	c.793+3A>C	r.742_793del (exon 6 skip)	p.(Pro248Ilefs * 15)
	*TMC8*	rs1381151589	NC_000017.11 (NM_152468.5)	g.78132031 G>A	c.298+1G>A	r.150_298del (exon 3 skip)	p.(Gln51Profs * 42)
Multiple exon skipping	*TSC2*	NA	NC_000016.10 (NM_000548.5)	g.2056989_ 2074645del	NA	r.775_2545del (exon 9 to 22 skip)	p.(Met260Trpfs * 44)
Alternative 5′ and 3′ splicing	*ENG*	NA	NC_000009.12 (NM_000118.4)	g.127819450 G>C	c.1311+172 C>G	r.1311_1312ins1311+1_1311+167	p.(Lys438Valfs * 8)
	*MED12* *	rs1556334519	NC_000023.11 (NM_005120.3)	g.71121602 G>A	c.887G>A	r.[887g>a;847_888del]	p.[Arg296Gln;Tyr283_Arg296del]

* This variant causes both a missense and a splice variant. NA; not applicable. RNA levels were confirmed by rLAS. Protein levels are based on the assumption that they are translated.

**Table 2 ijms-26-03220-t002:** Primer list.

Gene	Size (kb)	Forward Primer	Reverse Primer
TSC1	8.5	gtgctgtacgtccaagatgg	tagtgctttcagcgagaaaagg
TSC2	5.5	gggaggggttttctggtg	ctgacaggcaataccgtccaag
ENG	2.5	acaagtcttgcagaaacagtcc	acaagtcttgcagaaacagtcc
TMC8	2.3	tgcacagaggccatagccaa	gtaaggaggcctgaaggggc
MED12	6.7	gtcgagagtttctaacgtgcc	gggaattaagaggaaagggtgg
FH	1.7	cagaaattctacccaagctccc	acttgtttaatccatcttagacctagc
EDA	1.2	tcaagagagtgggtgtctcc	caacaccaatacacctcactcc

## Data Availability

The sequencing data presented in this study are available upon request from the corresponding author (H.U.), as they are subject to disclosure restrictions under the Japanese government’s Personal Information Protection Act, and the consent of the subjects was not obtained.

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
