# Peer review of "Computational Comparison of Differential Splicing Tools for Targeted RNA Long-Amplicon Sequencing (rLAS)"

_ijms, 2025, doi:10.3390/ijms26073220_

Round 1
Reviewer 1 Report
Comments and Suggestions for Authors
The authors mention that alternative splicing is a very important event for post-transcriptional regulation since multiple RNAs can be generated with changes in the sequence of the messenger RNA such as: exon skipping, multiple exon skipping, alternative 5’ splicing and alternative 3’ splicing among others. Nowadays, massive RNA seq analyses make it possible to identify these events, which is why it is necessary to use various analysis tools that help identify splicing events. Therefore, the objective of the study is to compare two tools HISAT2 and STAR, combined with MAJIQ, rMATS, MISO and SplAdder.
In my opinion, although they do a good approach on already known alterations (FH, EDA, TMC8, TSC2, ENG and MED12) and can support their findings in the mapping of splicing events. I do not perceive the originality of the work and possibly this manuscript is more appropriate for a journal focused on bioinformatics analysis.
Although they do describe the bioinformatics tools and the detection they can perform on splicing events, I think they could complement their manuscript with a transcriptome analysis, since the real challenge is to be able to identify splicing events in an RNA-seq analysis, since depth and coverage are essential.
Author Response
Dear Reviewer,
Thank you for giving us the opportunity to submit a revised draft of our manuscript titled “Computational comparison of differential splicing tools for targeted RNA long-amplicon sequencing (rLAS)” to the International Journal of Molecular Sciences. We appreciate the time and effort that you have dedicated to providing your valuable feedback on our manuscript. We are grateful to you for your insightful comments on our paper. We have been able to incorporate changes to reflect all of your suggestions. We highlighted the changes within the manuscript.
The rLAS method is more suitable for genetic diagnosis. We have revised this point (Line 60-64 in Page 2 and Line 267-270 in Page 9).
We look forward to hearing from you in due time regarding our submission and to respond to any further questions and comments you may have.
Sincerely,
Hiroki Ura, Ph.D.
Center for Clinical Genomics
Kanazawa Medical University Hospital
- Daigaku, Uchinada, Kahoku, Ishikawa, 920-0293, JAPAN
Phone No: +81 076-286-2211
Email Address: h-ura@kanazawa-med.ac.jp

Reviewer 2 Report
Comments and Suggestions for Authors
This Manuscript provides valuable insight into the strengths and weaknesses of different splicing tools for detecting alternative splicing events. MAJIQ demonstrated superior performance in identifying a wide range of splicing events but showed a minor limitation in missing one exon skipping event. Conversely, rMATS excelled in detecting exon-skipping events but struggled with other splicing variations, limiting its broader applicability. The comparison effectively highlights the differential capabilities of these tools. Still, the study would benefit from a deeper exploration of why these differences occur and whether combining the tools could improve overall detection accuracy. Additionally, evaluating more splicing detection tools could provide a more comprehensive understanding of their relative strengths and weaknesses in rLAS.
Is it a complete Exon skipping in TMC 8? There are signals in Shashmi plots for the exon and a part of intron 2. These signals could indicate potential splicing events or other biological processes. What does it mean biologically? Please correct the Figure 3 legend (H). Is the depth related to size and reads mapped, so how is that figure significant?
Isn’t it noticeable that the exon size is inversely proportional to mapping and becomes less mappable with STAR? Figures 4 and 5. The study compares HISAT2 and STAR but does not explore other splicing-aware aligners (e.g., GSNAP, SpliceMap, or minimap2 for long reads). These other aligners may have different biases and performance variations across different splicing events, which could introduce biases in splicing analysis. Additionally, the performance variation across different splicing events suggests that the choice of aligner may introduce biases in splicing analysis.
Does the ability of MAJIQ’s inability to detect one exon skipping depend on the spacing of introns? The detection performance varies across different splicing events. While MAJIQ and rMATS performed well, they still missed certain events. This suggests that the method might not be universally reliable for all splicing types, requiring further optimization. A broader benchmark with additional tools (e.g., SUPPA, Whippet) is crucial to provide a more comprehensive assessment of authors. Please explain.
The study does not mention the use of biological replicates. The conclusions drawn may be susceptible to sample-specific biases and should be validated across multiple independent datasets. Please discuss.
It’s highly recommended that the paper conclude with a solid conclusion. This will provide a clear and concise ending to the study, helping the audience understand the key takeaways. A table of read depth, the preferred Mapping tool, and a splice variant detection tool would further enhance the study.
Author Response
Dear Reviewer,
Thank you for giving us the opportunity to submit a revised draft of our manuscript titled “Computational comparison of differential splicing tools for targeted RNA long-amplicon sequencing (rLAS)” to the International Journal of Molecular Sciences. We appreciate the time and effort that you have dedicated to providing your valuable feedback on our manuscript. We are grateful to you for your insightful comments on our paper. We have been able to incorporate changes to reflect all of your suggestions. We highlighted the changes within the manuscript.
Here is a point-by point response to your comments and concerns.
Point 1: Is it a complete Exon skipping in TMC 8? There are signals in Shashmi plots for the exon and a part of intron 2. These signals could indicate potential splicing events or other biological processes. What does it mean biologically? Please correct the Figure 3 legend (H). Is the depth related to size and reads mapped, so how is that figure significant?
Response 1: It is not complete Exon skipping in TMC8 because the PSI of the Exon skipping is about 0.1. Due to the minimum junction coverage setting in Sashimi plot, the junction in this Exon skipping is not shown. We have revised this point (Line 156 in Page 5 and Line 182-183 in Page 6 and Line 216 in Page 7) and have corrected the Figure 3 legend (H).
Point 2: Isn’t it noticeable that the exon size is inversely proportional to mapping and becomes less mappable with STAR? Figures 4 and 5. The study compares HISAT2 and STAR but does not explore other splicing-aware aligners (e.g., GSNAP, SpliceMap, or minimap2 for long reads). These other aligners may have different biases and performance variations across different splicing events, which could introduce biases in splicing analysis. Additionally, the performance variation across different splicing events suggests that the choice of aligner may introduce biases in splicing analysis.
Response 2: The mapping rate were calculated in the specific gene region not but specific exons where the known splicing event was detected. HISAT2 tended to map the off-target region. The mapping software (HISAT2 and STAR) have different biases and performance variations. We have revised to emphasize this point (Line 294-299 in Page 9).
Point 3: Does the ability of MAJIQ’s inability to detect one exon skipping depend on the spacing of introns? The detection performance varies across different splicing events. While MAJIQ and rMATS performed well, they still missed certain events. This suggests that the method might not be universally reliable for all splicing types, requiring further optimization. A broader benchmark with additional tools (e.g., SUPPA, Whippet) is crucial to provide a more comprehensive assessment of authors. Please explain.
Response 3: The splicing software (MAJIQ and rMATS) also have different biases and performance variations. We have revised to emphasize this point (Line 294-301 in Page 9).
Point 4: The study does not mention the use of biological replicates. The conclusions drawn may be susceptible to sample-specific biases and should be validated across multiple independent datasets. Please discuss.
Response 4: In this study, the samples from the 7 patients with different known splicing events were used for rLAS. We have revised to emphasize this point (Line 301-304 in Page 9).
Point 5: It’s highly recommended that the paper conclude with a solid conclusion. This will provide a clear and concise ending to the study, helping the audience understand the key takeaways. A table of read depth, the preferred Mapping tool, and a splice variant detection tool would further enhance the study.
Response 5: We have revised this point (Figure 6).
We look forward to hearing from you in due time regarding our submission and to respond to any further questions and comments you may have.
Sincerely,
Hiroki Ura, Ph.D.
Center for Clinical Genomics
Kanazawa Medical University Hospital
- Daigaku, Uchinada, Kahoku, Ishikawa, 920-0293, JAPAN
Phone No: +81 076-286-2211
Email Address: h-ura@kanazawa-med.ac.jp

Reviewer 3 Report
Comments and Suggestions for Authors
The manuscript is well-written and ready for publish. My only suggestion is to add long-read sequencing (PacBio or Nanopore) in the Discussion.
Author Response
Dear Reviewer,
Thank you for giving us the opportunity to submit a revised draft of our manuscript titled “Computational comparison of differential splicing tools for targeted RNA long-amplicon sequencing (rLAS)” to the International Journal of Molecular Sciences. We appreciate the time and effort that you have dedicated to providing your valuable feedback on our manuscript. We are grateful to you for your insightful comments on our paper. We have been able to incorporate changes to reflect all of your suggestions. We highlighted the changes within the manuscript.
We have added the long-read sequencer such as PacBio and Nanopore in the Discussion (Line 305-307 in Page 9 and Line 308-309 in Page 10).
We look forward to hearing from you in due time regarding our submission and to respond to any further questions and comments you may have.
Sincerely,
Hiroki Ura, Ph.D.
Center for Clinical Genomics
Kanazawa Medical University Hospital
- Daigaku, Uchinada, Kahoku, Ishikawa, 920-0293, JAPAN
Phone No: +81 076-286-2211
Email Address: h-ura@kanazawa-med.ac.jp

Round 2
Reviewer 1 Report
Comments and Suggestions for Authors
I think it's an interesting tool for detecting co-expressed splicing variants in different diseases. However, the approach has been directed at known splicing sites. In my opinion, RNA-seq analysis (Oxford Nanopore or PacBIO) would be necessary. The approach is based on amplicon sequencing, and in this context, PCR favors the identification of multiple co-expressed transcripts. In contrast, RNA-seq analysis, where amplification is not involved, could limit the identification of these variants. Furthermore, their results mention that 250 reads are required for a variant to be considered in the rLAS model. This seems to me to be a limitation because RNA-seq, due to the representativeness of these variants, may miss many.
Author Response
We are grateful to you for your insightful comments on our paper. We have been able to incorporate changes to reflect all of your suggestions. We highlighted the changes within the manuscript.
The rLAS method is more suitable for genetic diagnosis. We have revised this point (Line 309-310 in Page 9 and Line 311-318 in Page 10). As your suggestion, it is better to analyze using long-read sequencer such as Nanopore and PacBio. The rLAS method provides the low-cost genetic diagnosis by focusing on the target disease-associated genes.
We look forward to hearing from you in due time regarding our submission and to respond to any further questions and comments you may have.
Sincerely,
Hiroki Ura, Ph.D.
Center for Clinical Genomics
Kanazawa Medical University Hospital
- Daigaku, Uchinada, Kahoku, Ishikawa, 920-0293, JAPAN
Phone No: +81 076-286-2211
Email Address: h-ura@kanazawa-med.ac.jp

Round 3
Reviewer 1 Report
Comments and Suggestions for Authors
The scope and limitations of the study are clear in the manuscript. I have no objection to the manuscript being accepted.
Comments on the Quality of English LanguageThe scope and limitations of the study are clear in the manuscript. I have no objection to the manuscript being accepted.